

# Sensitivity of atmospheric CO₂ to regional variability in particulate organic matter remineralization depths

Jamie D. Wilson[1,2], Stephen Barker[2], Neil R. Edwards[3], Philip B. Holden[3], and Andy Ridgwell[1,4]

[1]BRIDGE, School of Geographical Sciences, University of Bristol, UK
[2]School of Earth and Ocean Sciences, Cardiff University, UK
[3]School of Environment, Earth and Ecosystems, Open University, UK
[4]Department of Earth Sciences, University of California, Riverside, USA

**Correspondence:** Jamie D. Wilson (jamie.wilson@bristol.ac.uk)

**Abstract.**

The concentration of $CO_2$ in the atmosphere is sensitive to changes in the depth at which sinking particulate organic matter is remineralised: often described as a change in the exponent "$b$" of the Martin curve. Sediment trap observations from deep and intermediate depths suggest there is a spatially heterogeneous pattern of b, particularly varying with latitude, but disagree
over the exact spatial patterns. Here we use a biogeochemical model of the phosphorus cycle coupled with a steady-state representation of ocean circulation to explore the sensitivity of preformed phosphate and atmospheric $CO_2$ to spatial variability in remineralisation depths. A Latin hypercube sampling method is used to simultaneously vary the Martin curve indepedently within 15 different regions, as a basis for a regression-based analysis used to derive a quantitative measure of sensitivity. Approximately 30% of the sensitivity of atmospheric $CO_2$ to changes in remineralisation depths is driven by changes in
the Subantarctic region (36°S to 60°S), simliar in magnitude to the Pacific basin despite the much smaller area and lower productivity. Overall, the absolute magnitude of sensitivity is controlled by export production but the relative spatial patterns in sensitivity are predominantly constrained by ocean circulation pathways. The high sensitivity in the Subantarctic regions is driven by a combination of high export production and the high connectivity of these regions to regions important for the export of preformed nutrients such as the Southern Ocean and North Atlantic. Overall, regionally varying remineralisation
depths contribute to variability in $CO_2$ of between ± 5 - 15 ppm relative to a global mean change in remineralisation depth. Future changes in the environmental and ecological drivers of remineralisation, such as temperature and ocean acidification, are expected to be most significant in the high latitudes where $CO_2$ sensitivity to remineralisation is also highest. The importance of ocean circulation pathways to the high sensitivity in Subantarctic regions also has significance for past climates given the importance of circulation changes in the Southern Ocean.

# 1   Introduction

Sinking particles of organic matter transfer 5-10 Pg C per year from the upper ocean to the ocean interior (Henson et al., 2011), as part of a process known as the biological pump. As these particles sink, they are remineralised through bacterial and zooplankton-related activity, releasing the carbon and nutrients back into solution at depth. Vertical fluxes of particulate





organic carbon (POC) in the water column have historically been described by the Martin Curve, a power-law function that describes the rapid decrease in flux ($F_z$) from a maximum value at depth $z_0$, nominally the base of the mixed layer, to a small asymptotic value in deep waters (equation 1: Martin et al., 1987) (Fig. 1):

$$F_z = F_{z_0} \left( \frac{z}{z_0} \right)^{-b} \tag{1}$$

5     The dimensionless exponent in the power-law ('$b$') describes whether organic matter is remineralised predominantly at shallower (larger values of $b$, e.g., $b$=1.6) or deeper depths (smaller values of $b$, e.g., $b$=0.4) (Fig. 1). The exponent itself parameterises the rate at which POC sinks through the water column (units of m day$^{-1}$) and the rate at which it is remineralised (units of day$^{-1}$) (Kriest and Oschlies, 2008; Lam et al., 2011). In this paper we use the term 'remineralisation depth', defined as the depth at which $\sim$63% of POC has been remineralised (Kwon et al., 2009), to refer to changes in POC remineralisation 10 as it is relatable to alternative mathematical functions also used (e.g., Cael and Bisson, 2018).

    Ocean biogeochemical models predict that the concentration of $CO_2$ in the atmosphere is sensitive to changes in a globally uniform remineralisation depth. Kwon et al. (2009) showed that a deepening of the remineralisation depth globally of 24 m (from $b$ =1.0 to 0.9), redistributed dissolved inorganic carbon (DIC) from the intermediate waters to the deep ocean leading to a reduction in atmospheric $CO_2$ of between 10 and 27 ppm. The drawdown was also associated with a decrease in the 15 global mean concentration of preformed nutrients in the ocean interior (nutrients that are not utilised by biology in the surface ocean and enter the ocean interior via circulation: Ito and Follows, 2005). Kwon et al. (2009) found that increase in respired carbon in the deep ocean was balanced by a reduction in preformed nutrients exported in the North Atlantic. Deepening of the POC remineralisation depth could also drive dissolution of calcium carbonate ($CaCO_3$) in ocean sediments ultimately drawing down more $CO_2$ over millennial timescales (Roth et al., 2014). The potential impact of remineralisation depth changes on 20 atmospheric $CO_2$ is therefore a highly relevant component of the marine carbon cycle for both past and current changes in climate (Riebesell et al., 2009; Hülse et al., 2017; Meyer et al., 2016).

    Analyses of global sediment trap observations suggest there is a spatially heterogeneous pattern of remineralisation depths in the modern ocean that varies particularly with latitude. A synthesis of observations from deep sediment traps (>1500–2000 m: Henson et al., 2012) suggests that POC fluxes in high latitudes attenuate faster with depth (shallower remineralisation depth: 25 $b$=1.6) than in low latitudes, where a greater proportion of POC is transported to depth (deeper remineralisation depth: $b$=0.4), (Fig. 1). However, POC fluxes measured using neutrally buoyant sediment traps at shallower depths (<1000 m) suggest the inverse of this latitudinal pattern (Marsay et al., 2015) (see also, Weber et al., 2016). A recent compilation of sediment trap data and profiles of particle size distributions observed in the water column highlight additional intra-basin variability in $b$ (e.g., shallower remineralisation in the East Equatorial Pacific than in the West) and inter-basin variability (e.g., deeper 30 remineralisation in the Atlantic and Indian basins compared to the Pacific) (Guidi et al., 2015). The uncertainty in the spatial variability of remineralisation depths presents a challenge for determining which mechanisms may be responsible for changes in remineralisation depths and how these might drive future or past changes in remineralisation (e.g., Boyd, 2015). Additionally, this also presents a challenge for biogeochemical models that are beginning to resolve the mechanisms that are potentially





responsible for these spatial patterns such as particle size dependent sinking rates (DeVries et al., 2014), temperature dependent remineralisation (John et al., 2014), and oxygen dependence (Laufkötter et al., 2017).

A key question in light of the observed spatial variability in remineralisation depths and the associated uncertainty in spatial patterns is: what is the sensitivity of atmospheric $CO_2$ concentrations to spatial variability in remineralisation depths? Kwon
et al. (2009) further quantified the sensitivity of atmospheric $CO_2$ to basin scale changes in remineralisation depths by perturbing them in each basin individually, finding that the Pacific, Southern Ocean (defined as $>40°S$), Atlantic and Indian Oceans contributed 38%, 22%, 21% and 19% of the total $CO_2$ drawdown respectively (Kwon et al., 2009). The variability in $CO_2$ sensitivity between basins matched the variability in the magnitude of export production integrated over the basins and basin area, suggesting that no one region was more significant when varying the globally uniform remineralisation depth (Kwon
et al., 2009). However, this basin-scale analysis does not resolve the sensitivity of atmospheric $CO_2$ occuring at the resolution suggested by observations, i.e., a latitudinal and within-basin scale, or at the resolution of ecological and biogeochemical variability (Longhurst, 1998; Fay and McKinley, 2014). Additionally the analysis does not allow for the identification of potential interactions and feedbacks between regions when remineralisation depths are changing simultaneously.

Here we aim to address these issues by performing a global sensitivity analysis of regionally varying remineralisation depths.
To this end, we use the $2.8°$ resolution MITgcm transport matrix (a steady-state computationally efficient representation of ocean transport) with a model of phosphorus cycling where the ocean is divided into 15 regions in which remineralisation depths can change independently. Remineralisation depths are perturbed simultaneously using Latin hypercube sampling and sensitivity quantified using regression analysis, and related to changes in atmospheric $CO_2$ via preformed nutrients.

## 2   Methods

### 20   2.1   Model Description

We provide a brief description of the model here and a more detailed description in Appendix A. The approach to quantifying sensitivity used here relies on the ability to run an ensemble of model experiments. To make this approach feasible we use the 'transport matrix method' (Khatiwala et al., 2005; Khatiwala, 2007), a steady-state computationally efficient representation of ocean transport derived from a dynamic ocean model. We use monthly mean transport matrices derived from the $2.8°$ global
configuration of the MIT ocean model with 15 vertical levels (Khatiwala et al., 2005; Khatiwala, 2007). These specific matrices have been previously applied to model biogeochemistry (Kriest et al., 2012; Kriest and Oschlies, 2015).

The biogeochemical model used here is a model of the marine phosphorus cycle that resolves phosphate ($[PO_4]$) and dissolved organic phosphorus (DOP), similar to other models used to quantify the sensitivity of the biological pump (DeVries et al., 2012, 2014; Pasquier and Holzer, 2016). Following Kwon et al. (2009), we calculate the production of organic mat-
ter using either a nutrient-restoring scheme, $[PO_4]$ is restored to monthly observations of $[PO_4]$ (Garcia et al., 2014) with a timescale of 30 days (eqn. A3), and one with constant export production where export production is fixed to that of the control run unless nutrients fall below zero. These two schemes represent two end-member scenarios where organic matter production either depends entirely on macronutrient concentrations and can increase with higher nutrient fluxes (restoring) or is limited by

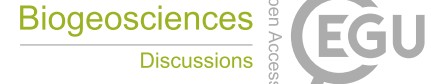



other factors such as light or micronutrients (constant export). The remineralisation of particulate organic phosphorus (POP) is parameterised using the Martin curve (eqn 1). We use preformed $PO_4$ ($PO_4^{pre}$) to relate changes in the modelled phosphorus cycle to changes in atmospheric $CO_2$ using a statistical relationship derived from published experiments (Appendix B). This provides a way of relating changes in our model of phosphorous to changes in atmospheric $CO_2$ without simulating a relatively

computationally expensive carbon cycle.

## 2.2 Experiment Design

### 2.2.1 Defining Regions

We define a set of oceanic regions to approximately encapsulate the large-scale variability in biogeochemistry and patterns of remineralisation depths observed in sediment trap studies. We define regions by lines of latitude and basins, similar to the

approach used by air-sea flux inversion studies, e.g., (Gloor et al., 2001; Mikaloff Fletcher et al., 2006). 15 regions are defined based on a partitioning by Gloor et al. (2001) with some minor changes (Fig. 2a). The assigned regions broadly correspond with major features in observed surface $[PO_4]$ such as higher concentrations in upwelling regions and lower concentrations in the nutrient-depleted gyres (Fig. 2b). This suggests the regions should be a reasonable analogue for an alternative approaches that capture key spatial variability in ecology and biogeochemistry by defining regions using vertical mixing, mixed layer depths,

sea ice and sea surface temperature (Longhurst, 1998; Sarmiento et al., 2004; Henson et al., 2010; Fay and McKinley, 2014). The regions are also comparable to the ocean biomes defined used in previous biological pump studies (e.g., Weber et al., 2016; Pasquier and Holzer, 2016).

## 2.3 Sensitivity Analysis

We first perform a set of reference experiments where the Martin curve exponents are varied between 0.4 and 1.6 globally, i.e.,

all regions are assigned the same Martin curve exponent. The range is based on the range of spatial variability observed in the modern ocean (Henson et al., 2012; Marsay et al., 2015; Guidi et al., 2015). Each experiment is run for 10,000 years from initial uniform conditions using the nutrient-restoring scheme to predict export production (eqn. A3). We define the control run as the experiment with the lowest root mean square misfit compared to annual mean World Ocean Atlas 2013 $[PO_4]$ observations (Garcia et al., 2014). We find the lowest misfit when $b$=1.0 globally, as found in other studies using the same MITgcm transport

matrices (Kriest et al., 2012). A second set of reference experiments are then run with a constant-export scheme using the export production from the control run.

  We perform a global sensitivity analysis with the aim to quantitatively rank the sensitivity of atmospheric $CO_2$ to remineralisation depth changes in each region (e.g., Pianosi et al., 2016). Latin hypercube sampling, a stratified-random procedure that provides an efficient way of sampling high dimensional parameter space (McKay et al., 1979), is used to vary the Martin curves

in every region simultaneously for the global sensitivity analysis. Values of $b$ are sampled from a uniform distribution ranging from 0.4 to 1.6 using the 'lhsdesign' function in MATLAB with 'maximin' sampling. The range of $b$ used centres around $b$=1.0 as used for the control run. We generate a Latin hypercube ensemble with 200 experiments, balancing the need for higher



sampling resolution of the parameter space and total computational time. We run two sets of the Latin hypercube experiments: one with nutrient-restoring export production and the other with constant export production where export production is taken from the control run. All experiments are run for 3000 years following on from the control run which is sufficient for the mean deep ocean $[PO_4]$ to equilibrate to a global change in the Martin Curve (Kwon et al., 2009). Annual mean fields of $[PO_4]$ are

diagnosed from the last full simulation year.

We use multiple linear regression analysis to derive the sensitivity of atmospheric $CO_2$ to changes in $b$ in each region ($k$) where the fitted coefficients ($\beta_k$) give a quantitative measure of the sensitivity (e.g., Pianosi et al., 2016):

$$CO_2 = \beta_0 + \sum_k \beta_k b_k \tag{2}$$

## 3   Results

### 3.1   Sensitivity of $CO_2$ to regional variability in remineralization depths

To quantify the sensitivity of $CO_2$ to regional changes in $b$ we fit linear regression models to the results of the Latin hypercube ensembles. The resulting regression models explain a large proportion of the variability between $CO_2$ and $b$ ($R^2$ = 0.88 and 0.90 for the constant-export and restoring-uptake ensembles respectively). Residuals of the regression models showed no significant bias versus the regression output (not shown) suggesting that a linear model was appropriate. Although the relationship

between $CO_2$ and a globally-uniform remineralisation depth is non-linear (e.g., Fig. 6), the relationship is near linear around the observed global mean in the centre of the range of $b$ tested (see also, Kwon et al., 2009). Overall, the absence of a strongly non-linear relationship suggests the use of a linear regression model is appropriate (Pianosi et al., 2016).

When $b$ is varied as a globally uniform parameter from 0.4 to 1.6, atmospheric $CO_2$ varies from 219 to 303 ppm (range of 84 ppm) for the constant-export scheme and from 263 to 284 ppm (range of 21 ppm) for the nutrient-restoring scheme, consistent

with previous model experiments (Kwon et al., 2009). Figure 3 shows how the sensitivity of $CO_2$ to changes in $b$ varies as a function of region. $CO_2$ is most sensitive to changes in $b$ occuring in the Subantarctic regions, with $CO_2$ being most sensitive to changes in the Indian sector of the Subantarctic (Fig. 3, Table 1). The Southern Ocean and sub-tropical gyres, with the exception of the gyre in the North Pacific, are consistently the regions where $b$ has the smallest impact on $CO_2$. Other regions, including the equatorial Indian ocean, Equatorial Pacific and North Pacific have an intermediate sensitivity. As a region, the

Subantarctic is responsible for ∼30% of the $CO_2$ sensitivity, comparable to the Pacific basin-scale sensitivity (Table 1).

As with the globally uniform changes in $b$, the magnitudes of regional sensitivities are smaller when run with nutrient-restoring uptake as opposed to a constant-export scheme because export production is able to convert any changes in surface nutrient fluxes back into organic matter, limiting any change in preformed nutrients. However, the relative sensitivity ranked across regions remains similar, as shown by expressing $b_k$ as a percentage (Table 1). Therefore, the regional patterns in Figure

3 are not sensitive to assumptions about the response of nutrient uptake to the redistribution of nutrients. This suggests the



absolute magnitude of $CO_2$ sensitivity to changes in $b$ is related to global export production that is is not driven by local changes in export production specific to any region(s).

Kwon et al. (2009) demonstrated that the sensitivity of $CO_2$ to basin-scale changes in $b$ correlated with the magnitude of export production in each basin. Similarly, we find a general postitive relationship between sensitivity and regional export

production, as measured by the mean annual average export production across the 200 ensemble runs (Fig 4). Intuitively, regions with lower export production, i.e., that contribute less to the inventory of regenerated $PO_4$, have a smaller impact on the balance between preformed and regenerated nutrients and therefore on atmospheric $CO_2$. Whilst $CO_2$ is generally more sensitive to remineralisation depths in regions with higher export production, sensitivity varies across regions with similar export productivity. For example, the sensitivity for the temperate North Pacific (NTemp-PAC; Fig. 4a) ($\Delta CO_2/\Delta b = 15.0$,

export production = $1.4\pm0.11$ Tmol P year$^{-1}$) is approximately double that of the sub-polar region of the Southern Pacific (STemp-PAC; Fig. 4a) ($\Delta CO_2/\Delta b = 6.3$, export production = $1.4\pm0.17$ Tmol P year$^{-1}$). There are no apparent relationships between the variability of export productivity across the ensemble in each region, as shown by the horizontal errorbars, and sensitivity (Fig. 4). This further supports the finding that the response of export production to changes in nutrient distributions are not an important factor in the sensitivity of $CO_2$ to regional changes in $b$.

The variability in sensitivity not explained by the magnitude of POC export is likely a function of how changing remineralisation depths interact with ocean circulation to redistribute nutrients. To quantify this effect we calculate $[\overline{PO_4^{pre}}]$ in the ocean interior derived from each region individually (see Appendix B) and repeat the sensitivity regression analysis:

$$\overline{PO_4^{pre}} = \beta_0 + \sum_k \beta_k b_k \qquad (3)$$

The new regression analysis (eqn. 3) now predicts the contribution of changing $b$ in all regions to the change in $[\overline{PO_4^{pre}}]$

in single region rather than globally (Fig. 5). $PO_4^{pre}$ is normalised prior to the regression to make the coefficients comparable between regions which otherwise vary by up to three orders of magnitude. As such, Figure 5 shows the relative sensitivity of $[PO_4^{pre}]$ to changes in $b$. $R^2$ ranges from 0.82 to 0.97 suggesting overall the linear regression models are appropriate. The regression coefficients specific to a single region are collected from across the 15 regression results in each panel of Figure 5 to show the relative sensitivity of $PO_4^{pre}$ exported across all regions in response to changes in $b$ local to the region corresponding

with panel. For example, the Southern Ocean panel shows how a change in $b$ in the Southern Ocean affects $PO_4^{pre}$ in all other regions. The sensitivity locally is coloured in black and positive sensitivities indicate an increase in $[\overline{PO_4^{pre}}]$ as $b$ increases in value, i.e., shallower remineralisation. Figure 5 displays the results for the constant-export experiments whereas the equivalent results for the restoring-uptake experiments are shown in Figure S3.

The sensitivity analysis for each region on an individual basis shows that changes in $b$ in the Subantarctic regions have large

impacts on $PO_4^{pre}$ across regions globally (Fig. 5). In particular, these regions have a particular effect on the $PO_4^{pre}$ export in the Southern Ocean and in the Atlantic as a basin, comparable in magnitude to the local changes in $PO_4^{pre}$ (compare size of black bars to other bars: Fig. 5). Changes in $b$ in the equatorial upwelling regions of the Pacific and Indian Oceans also have a large global effect but with a more pronounced local effect. These features are more pronounced with nutrient-restoring uptake





(Fig. S3). The Southern Ocean and North Atlantic regions are those with the highest $PO_4^{pre}$ export and variability across the ensemble, consistent with previous findings about the global importance of these regions for preformed $PO_4$ (DeVries et al., 2012; Pasquier and Holzer, 2016). This suggests the larger sensitiviy of $CO_2$ to changes in $b$ in the Subantarctic regions is due to the way in which the ocean circulation connects these regions to the Southern Ocean and North Atlantic. In contrast,

changing $b$ in the Southern Ocean and North Atlantic has a relatively minimal effect on $PO_4^{pre}$ (and by inference $CO_2$) both locally and globally.

### 3.2 Regional versus Global Sensitivity

Lastly, we explore whether the spatial patterns in sensitivity (Fig. 3) are significant on a global scale. Global average values of $b$ are calculated for each of the 200 Latin hypercube samples using an area-weighted mean and compared against experiments

where $b$ is pertubed uniformly across regions (Fig. 6). We find that the relationship between $CO_2$ and global mean $b$ matches closely to that with the globally uniform $b$ but with an offset of $\sim$20 ppm and $\sim$10 ppm for the constant and restoring export schemes respectively (Fig. 6a). We suggest this offset is likely caused by a non-linear relationship between $b$ and the amount of organic matter reaching the deep ocean (as measured by the $e$-folding depth: depth at which $\sim$63% of exported POC has been remineralised) following from the fact that the Martin curve represents the scenario of a fixed remineralisation rate and an

increasing sinking rate (Kriest and Oschlies, 2008; Cael and Bisson, 2018) (see Supplementary Material). A change in $b$ from 1.4 to 1.3 results in a decrease in $e$-folding depth of 14 m whereas a change in $b$ from 0.4 to 0.3 results in a change of 1902 m. Therefore, larger values of $b$, i.e., shallower remineralisation, have disproportionally more weight when averaging spatially variable $b$ values. To demonstrate this, we find the equivalent $e$-folding depths for each Latin hypercube sample, which form a skewed distribution due to higher occurrence of shallower remineralisation, calculate the area-weighted geometric mean $e$-

folding depth for each sample, and re-arrange again for $b$ (see Supplementary Material for details). The distributions in Figure 6b now fall along the line of globally uniform experiments.

The relationship between $CO_2$ and the globally averaged $b$ values from the sensitivity experiments closely matches the relationship between $CO_2$ and globally uniform $b$ for the both constant-export and restoring uptake schemes (Fig. 6a). The average regionally varying $b$ values vary within $\sim \pm 15$ ppm of the globally uniform experiments with constant-export and

$\sim \pm 5$ ppm for the nutrient-restoring experiments, comparable to the change in $CO_2$ for a globally uniform change in $b$ of $\sim$0.2.

## 4 Discussion

Sediment trap observations reveal significant spatial variability in remineralisation depths. Here we have quantified the sensitivity of atmospheric $CO_2$ to regional changes in remineralisation depths and show that $CO_2$ is most sensitive to changes in the Subantarctic regions. Much of the observed spatial variability varies across latitudes (Henson et al., 2012; Guidi et al., 2015;

Marsay et al., 2015; Weber et al., 2016). Additionally, the mechanisms potentially driving these patterns are also likely to vary on a latitudinal basis, with changes in related environmental properties in response to anthropogenic $CO_2$ emissions affecting the high latitudes in particular: temperature changes (Kirtman et al., 2013) affecting temperature-dependent remineralisation




rates; a reduction in carbonate saturation state with ocean acidification (Orr et al., 2005) affecting ballasting and changes in plankton community composition; and cell size (Lefort et al., 2015) affecting aggregation dynamics and particle sinking velocities. Additionally, this is a consideration for changes in remineralisation depths occurring in past climates (Meyer et al., 2016). This suggests that the spatial patterns in $CO_2$ sensitivity could be significant when considering the impact of remineralisation

depth changes.

Changes in the air-sea balance of carbon are commonly related to changes in preformed nutrients. Because of the inefficient utilisation of upwelled nutrients in the Southern Ocean this region has been identified as key to setting the efficiency of the biological pump (Ito and Follows, 2005; DeVries et al., 2012). Our results show that this is also key for the the sensitivity of $CO_2$ to regional variability in remineralisation depths because of upwelling in the Subantarctic regions (Fig. 5). This re-

lationship has implications when invoking changes in the efficiency of the biological pump in past climates such as the Last Glacial Maximum (LGM). Processes that increase the utilisation of nutrients in the Southern Ocean, such as iron fertilisation, and processes that reduce the delivery of nutrients to the Southern Ocean, such as increased stratification, have been implicated in the drawdown of atmospheric $CO_2$ during the LGM (Sigman et al., 2010). Any changes in strafication will also impact the sensitivity of $CO_2$ to any additional changes in remineralisation depths, such as from changes in ballasting minerals and/or

temperature dependent remineralisation (Chikamoto et al., 2012). In comparison, processes such as iron fertilisation will not impact on this sensitivity. Because the spatial patterns of $CO_2$ sensitivity to regional changes in remineralisation are predominantly constrained by ocean circulation pathways, this also suggests that the sensitivity may change with a reorganisation of ocean circulation as suggested for the LGM (Sigman et al., 2010).

The Martin curve is a commonly used parameterisation of the remineralisation of particulate organic matter with depth in

marine biogeochemical models, and is commonly applied with a globally uniform exponent ($b$) (Hülse et al., 2017). However, the Martin curve used in this way has potential limitations: it is an empirical and static parameterisation that does not represent the mechanisms affecting remineralisation and sinking rates; and it does not capture spatial variability in remineralisation observed in sediment trap data (Henson et al., 2012; Marsay et al., 2015; Guidi et al., 2015). In our sensitivity analysis, we have shown that $CO_2$ has a similar sensitivity to the global mean change in $b$ as compared to a globally uniform change in $b$

with an uncertainty of $\pm 5$ - 15 ppm, equivalent to a change in $b$ of $\sim 0.2$ (Fig. 6). Kwon et al. (2009) suggest a decrease of 0.3 from the modern remineralisation depth is sufficient to explain the increase in deep ocean nutrient concentrations during the Last Glacial Maximum. For the 21st century, (Laufkötter et al., 2017) predict a decrease of POC export at 500 m by 2100 under RCP8.5 in response to temperature and oxygen-dependent remineralisation, equivalent to a decrease in $b$ of $\sim 0.25$. As such, the global mean change in potential future and past changes in remineralisation depth may be larger than the uncertainty

associated with spatial variability. This has potentially useful implications for modelling the remineralisation of particulate organic matter fluxes. Models resolving the various processes that affect remineralisation rates and sinking velocities have recently been developed (Jokulsdottir and Archer, 2016; Cram et al., 2018) however, the requirements to model processes such as particle aggregation can be computationally expensive, limiting their application to 1-D models (Jokulsdottir and Archer, 2016; Cram et al., 2018) or to offline models (DeVries et al., 2014). A globally uniform change in $b$ informed by these models

could then used to calculate the impact on atmospheric $CO_2$ if the change in $b$ is greater than 0.2.





Our results are dependent on the use of transport matrices derived from one ocean circulation model. The model is commonly applied to study biogeochemistry which means that our results should be consistent with a number of existing studies (e.g., Kriest et al., 2012). In addition, our results have key similarites, including absolute and relative magnitudes of regional preformed $PO_4$ export, to other studies using alternative steady-state circulations (DeVries et al., 2012; Pasquier and Holzer, 2016). As such, our results should be broadly reproducible with other models. A disadvantage to using a steady-state circulation is that we cannot quantify impact of the $CO_2$-climate feedback on ocean circulation and atmospheric $CO_2$. Studies exploring the simultaneous effects of warming temperatures on circulation and biology in response to anthropogenic $CO_2$ emissions show that changes in circulation could be as important as biological changes (Cao and Zhang, 2017), (but see, Yamamoto et al., 2018). Quantifying the regional sensitivity with a dynamic ocean is therefore an important focus for future research. Lastly, our modelling approach uses an empirical relationship between preformed nutrients and $CO_2$ that enfolds the effects of processes such as air-sea gas exchange and the export of $CaCO_3$. Ratios of $CaCO_3$ to POC vary latitudinally, and could therefore modify our sensitivity results. Segschneider and Bendtsen (2013) found important feedbacks involving interactions between calcifiers and silicifiers in an marine ecosystem model when exploring temperature dependent remineralisation rates in the 21st Century. Future model experiments including a representation of plankton ecosystems would therefore help explore the impact of $CaCO_3$ export on regional sensitivity patterns.

## 5 Conclusions

We have presented a sensitivity analysis that quantifies the sensitivity of atmospheric $CO_2$ to regional variability in particulate organic carbon remineralisation depths. $CO_2$ is most sensitive to changes in remineralisation depths occurring in the Subantarctic regions, particularly the Indian Sector. As a whole, the Subantarctic regions have a sensitivity similar to that of the Pacific basin despite the smaller area and levels of productivity. Sensitivity patterns are in part a function of the magnitude of export production in each region and the physical circulation pathways specific to each region. Whilst the overall magnitude of $CO_2$ sensitivity to regional changes is dependent on the magnitude and response of export production to changes in nutrients, the relative spatial patterns in sensitivity are predominantly constrained by ocean circulation pathways. We also find that the regional variability adds $\pm 5$ - $15$ ppm uncertainty to global mean changes in remineralisation depths. The regional patterns in sensitivity could be significant if a number of processes that potentially drive changes in remineralisation depths, including temperature-dependent remineralisation rates and plankton community structure, vary predominantly in the high latitudes. However, this uncertainty is similar to the change in $CO_2$ for a globally uniform change in $b$ of $\sim 0.2$ meaning that larger changes in $b$ could be reliably approximated by a globally uniform $b$ as commonly used in biogeochemical models.





## Appendix A:  Model Description

The Latin hypercube sampling approach used relies on the ability to run an ensemble of model experiments. To make this approach feasible we use the 'transport matrix method' (Khatiwala et al., 2005; Khatiwala, 2007). The model is written in Fortran 90 and achieves $\sim$1500 years hour$^{-1}$ on a single core.

### A1   Steady-state Ocean Circulation Model

The matrix used here is the 2.8° global configuration of the MIT model with 15 vertical levels driven by seasonally cycling fluxes of momentum, heat, and freshwater, publicly available from http://kelvin.earth.ox.ac.uk/spk/Research/TMM/TransportMatrixConfigs Seasonally varying ocean circulation is calculated at each timestep by linearly interpolating between monthly mean matrices. An advantage of using transport matrices is that the timestep can be made longer to reduce computational expense (Khatiwala, 2007). Here we extend the circulation timestep to 3.8 days.

### A2   Biogeochemical Model

The biogeochemical model represents the cycle of phosphorus in the ocean with two dissolved tracers, PO$_4$ and dissolved organic phosphorus (DOP).. The biogeochemical model has the same timestep as the ocean circulation model (3.8 days). PO$_4$ and DOP are governed by the following equations:

$$\frac{\mathrm{dPO_4}}{\mathrm{d}t} = \mathbf{A}\mathrm{PO_4} - J_{\mathrm{up}} + J_{\mathrm{POP}} + J_{\mathrm{DOP}} \tag{A1}$$

$$\frac{\mathrm{dDOP}}{\mathrm{d}t} = \mathbf{A}\mathrm{DOP} + v \cdot J_{\mathrm{up}} - J_{\mathrm{DOP}} \tag{A2}$$

where $\mathbf{A}$ denotes the transport matrix calculation of ocean transport and $J$ denotes biogeochemical source/sink terms.

The uptake of PO$_4$ during production of organic matter occurs in the euphotic zone, here defined as the base of the upper two grid-boxes (120 m). Following Kwon et al. (2009) we calculate the production of organic matter using either a nutrient-restoring scheme or a constant-export scheme. The nutrient-restoring scheme restores surface concentrations of PO$_4$ to observed [PO$_4$] with a restoring timescale ($\tau$ = 30 days Najjar et al., 2007) and is scaled by the fraction of seaice present ($F_{seaice}$):

$$J_{\mathrm{up}} = \frac{1}{\tau}\max\Big(\big(\mathrm{PO_4} - \mathrm{PO_{4,obs}}\big),0\Big)\big(1 - F_{seaice}\big) \tag{A3}$$

Organic matter production in the constant-export scheme is fixed to that of the experiment defined as the control run unless surface [PO$_4$] is depleted below zero in which case $J_{\mathrm{up}}$ is set to zero at that timestep. The control run is defined as having the run with the lowest root mean square misfit compared to annual mean World Ocean Atlas [PO$_4$] observations.



A fixed fraction ($v$=0.66) of the organic matter production integrated across the upper two grid-boxes is routed directly to dissolved organic phosphorus (DOP) and remineralised back to $PO_4$ in a first-order reaction with decay rate $\kappa$ throughout the water column:

$$J_{\text{DOP}} = \kappa\text{DOP} \tag{A4}$$

5    The remaining fraction of organic matter production ($(1 - kappa$=0.34) is integrated across the upper two grid-boxes and exported as particulate organic phosphorus (POP) at the base of the euphotic zone (120 m). The remineralisation of POP is parameterised with the Martin Curve (Equation 1). POP that has reached the sediment is remineralised fully in the lowermost grid-box of the water column, maintaining a closed system with respect to [$PO_4$]. As such, there is no sediment component in this model.

## Appendix B:  Preformed PO$_4$ and atmospheric CO$_2$

Changes in atmospheric $CO_2$ due to changes in the biological pump can be directly related to the inventory or average concentration of preformed $PO_4$ ($PO_4^{\text{pre}}$) if total nutrient concentrations are conserved (Ito and Follows, 2005; Marinov et al., 2008). This provides a way of relating changes in our model of the phosphorous cycle to changes in atmospheric $CO_2$ without simulating a relatively computationally expensive carbon cycle. The distribution of annual mean [$PO_4^{\text{pre}}$] for each run is calculated by splitting the transport matrices into "interior" matrices ($\mathbf{A^I}$) and "exterior" matrices ($\mathbf{B}$) for both the explicit and implicit matrices (subscripts $e$ and $i$ respectively) (see Khatiwala, 2007). The annual mean surface [$PO_4$] from the end of a simulation is set as a boundary condition and solving for the interior distribution of $PO_4^{\text{pre}}$:

$$(\mathbf{A_i^I A_e^I - I})\mathbf{PO_4^{\text{pre}}} = (\mathbf{A_i^I B_e + B_i})\mathbf{PO_4} \tag{B1}$$

The global mean concentration of of $PO_4^{\text{pre}}$ ($\overline{[PO_4^{\text{pre}}]}$ ) is related to $CO_2$ using a empirical quadratic function (eqn. B2). The functon is derived from a compilation of published $PO_4^{\text{pre}}$ sensitivity experiments performed from three different models (Ito and Follows, 2005; Marinov et al., 2008; Kwon et al., 2009) using a non-linear least squares regression (Fig. A1). Although the inventories of $PO_4^{\text{pre}}$ are model dependent (Duteil et al., 2012), the changes in $\overline{[PO_4^{\text{pre}}]}$ and $CO_2$ relative to those of the control run show consistent trends across the different models. The resulting regression fit (details in Figure caption) is used to estimate changes in $CO_2$:

$$CO_2 = (\beta_1 \Delta PO_4^{\text{pre 2}} + \beta_2 \Delta PO_4^{\text{pre}} + \beta_3) + CO_2^{\text{ctrl}} \tag{B2}$$

25

*Author contributions.* JDW designed the experiments, developed the model code and ran the model. JDW prepared the manuscript with input from all co-authors.



*Competing interests.* The authors declare that they have no conflict of interest.

*Acknowledgements.* This work is based on work originally conducted as part of a PhD project (JDW) associated with the UK Ocean Acidification Research Programme (UKOARP), grant NE/H017240/1 to AR and SB. JDW and AR acknowledge support via the EU grant ERC-2013-Cog-617313. We would like to thank Andrew Yool for his comments on an earlier version of the manuscript. We also thank Samar

5    Khatiwala for making transport matrices freely available.





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





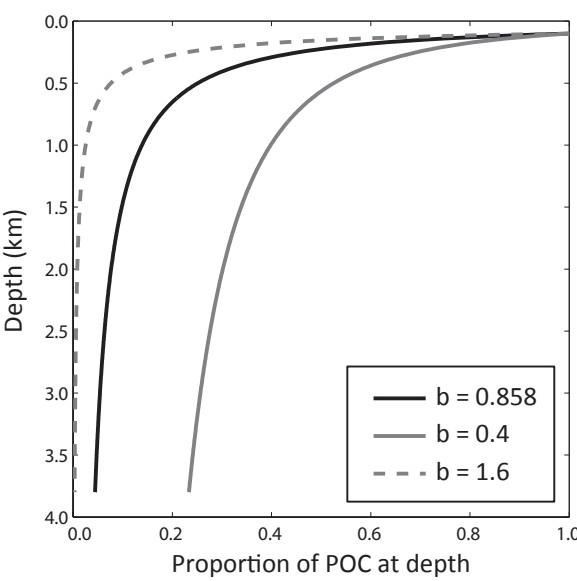

**Figure 1.** The normalised water column distribution of particulate fluxes defined using the Martin curve. As a comparison, Martin curves are shown with the exponent found by Martin et al. (1987) ($b$=0.858) and minimum and maximum exponent values used in this study based on sediment trap data compilations ($b$=0.4, $b$=1.6 Henson et al., 2012; Marsay et al., 2015; Guidi et al., 2015). All curves have export depth ($z_0$, eqn. 1) of 120 m.



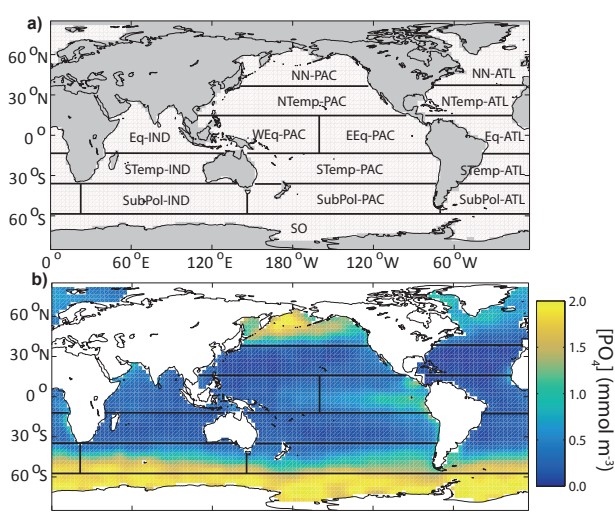

**Figure 2.** a) Location and names of the 15 regions defined on the model grid based on Gloor et al. (2001). Boundaries are at 58°S, 36°S, 13°S, 13°N, and 36°N. The equatorial Pacific is split at 98.75°E following Mikaloff Fletcher et al. (2006). Each region can be assigned a value of $b$ that is independent of other regions. b) Location of regions superimposed on the annual mean surface [PO$_4$] from World Ocean Atlas 13 (Garcia et al., 2014) regridded to the model grid.





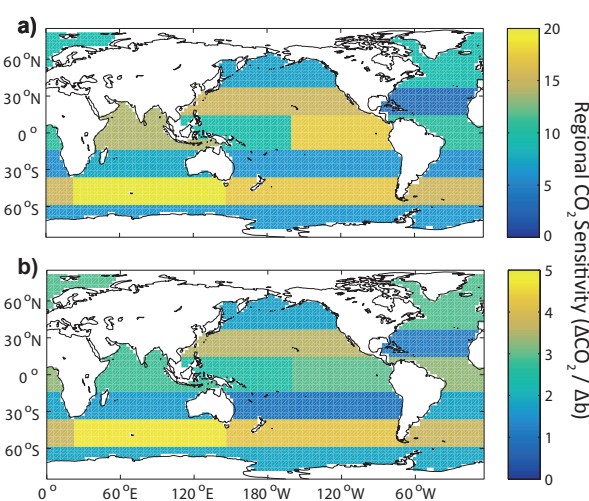

**Figure 3.** Regional sensitivity ($\Delta CO_2/\Delta b$) of atmospheric $CO_2$ (ppm) to changes in Martin curve exponents ($b$: unitless) for (a) the constant-export scheme and (b) the restoring-uptake scheme. A positive value relects an increase in $CO_2$ (preformed $PO_4$) with increasing $b$ (shallower remineralisation). Atmospheric $CO_2$ is inferred from modelled preformed $PO_4$ using the empirical relationship in Figure A1.





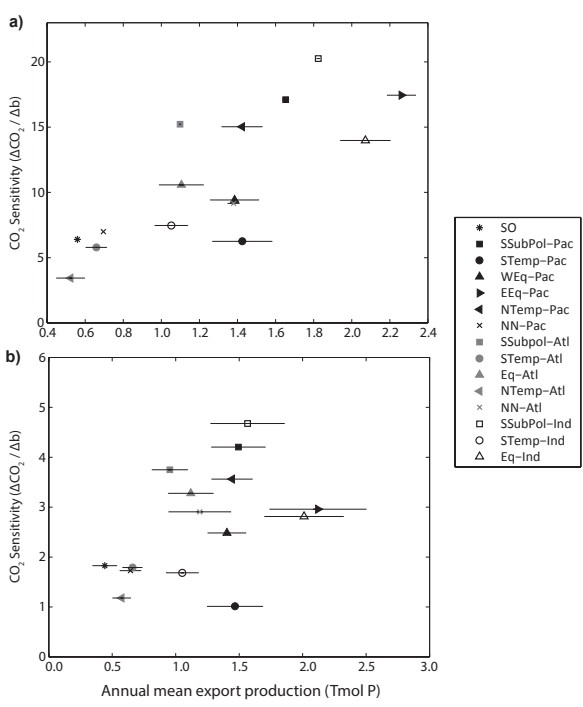

**Figure 4.** Relationship between regional $CO_2$ sensitivity and annual export of $PO_4$ in each region using (a) the constant-export scheme and (b) the restoring-uptake scheme. Annual export is shown as the mean of the 200 ensemble experiments with $\pm 1$ standard deviation error bars.



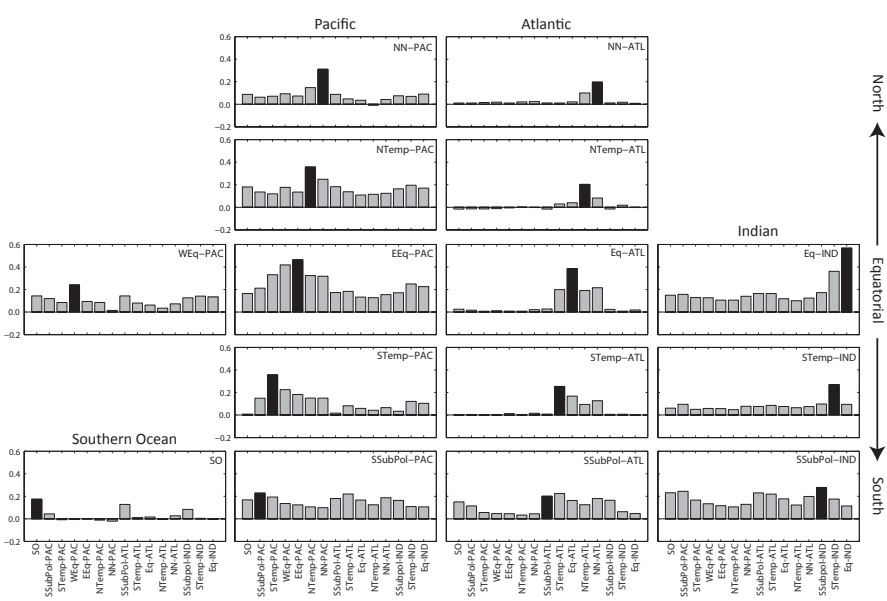

**Figure 5.** Normalised sensitivity of steady-state mean preformed [PO$_4$] in all regions to a local change in $b$ calculated with the constant-export scheme. Sensitivity is calculated using equation 3 and preformed [PO$_4$] is normalised before so that sensitivity can be compared on the same scale. The regression coefficients specific to a single region are collected from across the 15 regression results in each panel to show the relative sensitivity of PO$_4^{\text{pre}}$ exported across all regions (grey) in response to changes in $b$ local to the region (black) corresponding with panel. Panels are arranged by basins and by latitude. The equivalent plot for the restoring-uptake scheme is found in the supplementary material.





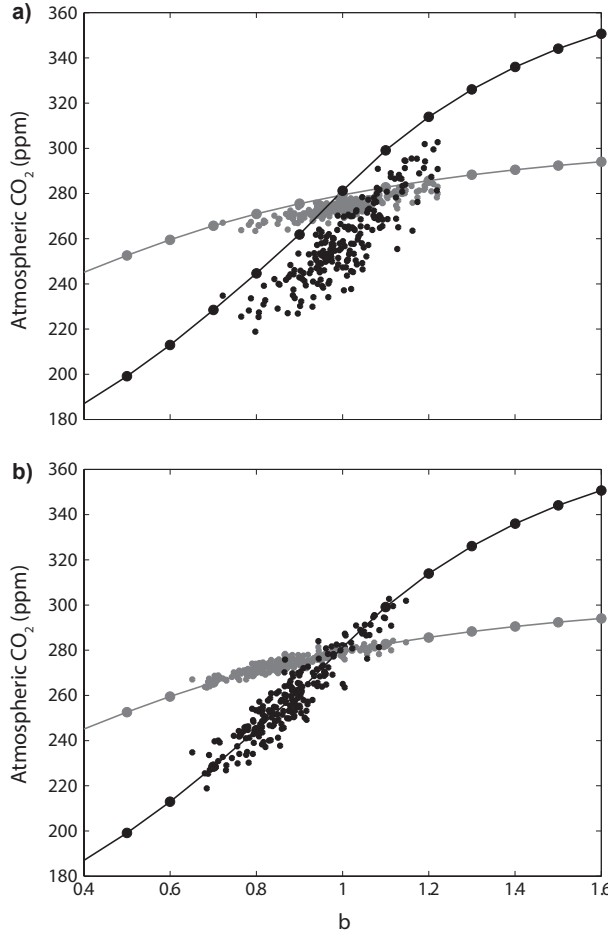

**Figure 6.** Comparison of $CO_2$ sensitivity when $b$ is varied as globally uniform parameter (solid lines) and when $b$ is varied regionally in the Latin hypercube samples and calculated as an (a) area-weighted global mean and (b) area-weighted geometric mean of $e$-folding depths converted back to $b$ to correct for non-linearities in the Martin Curve. Runs using the constant-export scheme are shown in black and restoring-uptake runs are shown in grey.





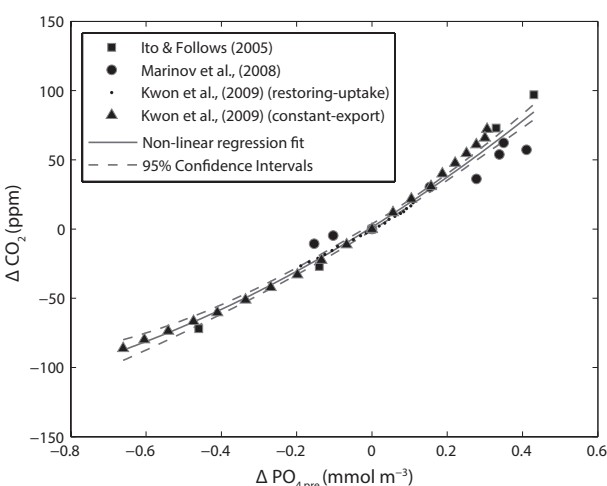

**Figure A1.** Relationship between relative changes in preformed $PO_4$ and atmospheric $CO_2$ from previously published model experiments (see legend for details). Relative changes are calculated by subtracting the respective values from the control run (for Marinov et al., 2008 this is defined as the LL - regular gas exchange run). All models feature gas exchange and no export production of $CaCO_3$. A quadratic function ($\Delta CO_2 = \beta_1 \Delta PO_{4,pre}^2 + \beta_2 \Delta PO_{4,pre} + \beta_3$) fitted to the combined data with non-linear least squares is shown with 95 % confidence intervals. The $R^2$ for the regression model is 0.97. The coefficients for the fit are $\beta_1$=54.12, $\beta_2$=170.15 and $\beta_3$=1.37





**Table 1.** Key metrics for each region including area, region-integrated mean annual POC export from the control run, $CO_2$ sensitivity for both the restoring-uptake and constant-export ensembles.

| | area (%) | POC Export (%)[a] | $\beta (\Delta CO_2/\Delta b)$ | $\beta$ (%) | $\beta (\Delta CO_2/\Delta b)$ | $\beta$ (%) |
|---|---|---|---|---|---|---|
| SO | 6.70 | 2.81 | 6.40 | 3.89 | 1.83 | 4.59 |
| SubPol-PAC | 7.67 | 8.35 | 17.11 | 10.40 | 4.20 | 10.55 |
| STemp-PAC | 10.37 | 7.86 | 6.25 | 3.80 | 1.01 | 2.54 |
| Weq-PAC | 7.21 | 7.46 | 9.42 | 5.72 | 2.48 | 6.23 |
| Eeq-PAC | 8.20 | 11.53 | 17.45 | 10.60 | 2.96 | 7.43 |
| Ntemp-PAC | 9.69 | 7.66 | 15.04 | 9.14 | 3.56 | 8.94 |
| NN-PAC | 4.94 | 3.50 | 6.99 | 4.25 | 1.73 | 4.34 |
| SubPol-ATL | 4.84 | 5.52 | 15.23 | 9.25 | 3.75 | 9.41 |
| Stemp-ATL | 4.46 | 3.53 | 5.79 | 3.52 | 1.79 | 4.49 |
| Eq-ATL | 5.45 | 6.06 | 10.58 | 6.43 | 3.28 | 8.22 |
| NTemp-ATL | 5.13 | 3.09 | 3.44 | 2.09 | 1.18 | 2.96 |
| NN-ATL | 4.84 | 7.04 | 9.16 | 5.57 | 2.91 | 7.29 |
| SubPol-IND | 6.86 | 9.18 | 20.26 | 12.31 | 4.68 | 11.73 |
| STemp-IND | 6.31 | 5.63 | 7.47 | 4.54 | 1.69 | 4.23 |
| Eq-IND | 7.32 | 10.85 | 13.98 | 8.49 | 2.81 | 7.06 |
| Subantarctic[b] | 19.38 | 23.04 | n/a | 31.96 | n/a | 31.69 |
| Pacific | 40.41 | 37.96 | n/a | 33.52 | n/a | 29.47 |
| Atlantic | 19.88 | 19.72 | n/a | 17.60 | n/a | 22.97 |
| Indian | 13.63 | 16.48 | n/a | 13.03 | n/a | 11.29 |

[a] POC export from the control run. [b] SubPol-PAC, SubPol-ATL and SubPol-IND