# Peer review of "Sensitivity of atmospheric CO₂ to regional variability in particulate organic matter remineralization depths"

_Biogeosciences, 2018_

## Referee Comment (RC1) · Anonymous Referee #1 · 11 Jan 2019

The authors explore the sensitivity of oceanic distribution of PO4 to global/regional changes in the remineralization depth, represented by the exponent of the Martin's powerlaw curve. The authors then infer the sensitivity of atmospheric pCO2 based on the previously published relationship between the oceanic inventory of preformed PO4 (or regenerated PO4) and atmospheric pCO2. I agree with the authors that the regional sensitivity can be an important component of the global carbon cycle response to climate change. Although the model experiments and analysis are sound and the paper is well written. I have some issues to be resolved.

Firstly, the model only infers the atmospheric CO2 response based on the published

relationship between the oceanic inventory change in preformed PO4 and the atmospheric CO2 change. The near-linear relationship between the two is only valid if we assume no solubility and air-sea CO2 disequilibrium effects. I acknowledge that the effects might be implicitly included in the empirical relationship extracted from a few previous global studies. However, I am not sure if the same relationship can be applied to the regional perturbation study. For example, any perturbations in the Southern Ocean (e.g., the ACC band where air-sea CO2 disequilibrium is large due to the short surface residence time of upwelled waters) might not lead to the atmospheric CO2 response proportional to the preformed PO4 response. This point can be especially worrisome because the most sensitive regions turn out to be the ACC band in the study.

Secondly, the sensitivity is estimated using the multiparameter linear regression method applied to the two sets of 200-member ensemble experiments where 15 regional exponents are perturbed simultaneously. Although the method seems sound, I wonder why the sensitivity should be quantified in this way? Are there any merits? Would the sensitivity be the same or different if the authors perturbed the exponent in a region at a time, requiring a total of 15 perturbation experiments for one model scheme? The individual perturbation experiments seem a simpler and cleaner way to quantify the atmospheric CO2 response to the perturbation and its relations with the export production in each domain.

Thirdly, the major novel finding is that the highest sensitivity in atmospheric CO2 is to the change in remineralization depth in Subantarctic regions due to high export production and the high connectivity to deep water formation regions. I see the reasoning behind it: The export production should be high because the export will determine how much regenerated PO4 can be affected by the perturbation. The connectivity to deep water formation regions is important because the deep water formation is the main pathway of preformed PO4 to the ocean's interior and the inventory of preformed PO4. However, I am not fully convinced by the authors' finding. Both "nutrient restoring" model and "constant export" model show that the sensitivity of atmospheric CO2 to the

remineralization depth change is also high in the "NTemp-PAC" domain (Fig. 3). Yet, the subtropical North Pacific is not a region with high export production nor close to any deep water formation regions. How can it be explained? Similarly, why are the deep water formation regions (i.e., the model NADW and AABW formation regions) not the sensitive regions?

It may be related to the third point. But I don't understand Figure 5. The sensitivity is normalized to what? What do the authors mean by "mean preformed PO4 in a region"? Is it the surface PO4 averaged over each region or the total preformed PO4 subducted from each region divided by the volume of water subducted from the same region?

---

## Referee Comment (RC2) · Anonymous Referee #2 · 21 Jan 2019

Wilson et al. present a sensitivity study of a simple global biogeochemical ocean model, in which they varied the particle flux exponent b globally and regionally, and investigate its effects on diagnosed atmospheric CO2 and global preformed phosphate in the ocean interior. Tracer transport is simulated using offline circulation matrices derived from the coarse-resolution MIT global model. Biogeochemistry is parameterised following Kwon et al. (2009), using a nutrient-restoring approach and a constant-export scenario. Their results show a large impact of b in the subantarctic region and western equatorial Pacific on CO2, and on preformed phosphate across regions. Compared to the effect of a global variation b (of ca. 80 ppm for b varying between 0.7-1.2), the local variation only adds some 5-15 ppm, i.e., seems to play a minor role.

[Figure]

I think this paper provides some very useful and new aspects regarding the importance of a potential regional variation of b, and its consideration in global biogeochemical models. Overall, the paper is well written, although I think the description of experimental design and analysis could be somehow improved. (I also noted some typos; see below.) My main concern is that the results are likely tied to the circulation model applied. As shown by Duteil et al. (2013; Biogeosciences, 10, 7723–7738, doi:10.5194/bg-10-7723-2013) the transport matrices from the MITgcm seem to suffer from far too large outcrop areas of dense waters in the Southern Ocean (their Fig. 2), indicating that the model circulation does not represent the real ocean in that region very well. Also, because of the very coarse resolution, this model might not represent the physical dynamics in the eastern equatorial Pacific very well. However, in the present study these two regions - the subantarctic regions and equatorial upwelling - have a large influence on $CO_2$ (Fig. 3 and 5). Thus, whereas this study provides important and interesting information for other global model studies that apply similar circulations (as noted in Discussion and Conclusions), I think that a few sentences on this are necessary to caution readers not familiar with the advantages and disadvantages of global circulation models. (To illustrate or investigate this point further, one could, e.g., look at the density distribution or mixed layer depths of the model.)

There seems to be a strong sensitivity of $CO_2$ to changes in b in the constant-export scheme (Fig 3), and also a clear relationship to export (Fig 4a). In contrast, normalized (by what?) preformed phosphate seems to be more sensitive in the nutrient restoring scenario (Fig 5 vs Fig S3), and no relationship seems to exist between $CO_2$ sensitivity and export (Fig 4b). I think these contrasting patterns for both model types deserve a bit more discussion. Perhaps some section plots of, e.g., density across the Pacific and Atlantic (see above) could aid the disussion about the effects of circulation vs. export type ("biogeochemistry"). If the circulation model is anywhere near the real world, some insight regarding the "connectivity" of different regions might perhaps be gained from the data-constrained analysis of water fractions presented by Khatiwala et al. (2012; Earth and Planetary Science Letters, 325–326, 116–125).

[Figure]

Specific comments:

p 3, line 15: "MITgcm" sounds like technical slang to me - is there a better word for it?

Section 2.3: At first, I had difficulties understanding the experimental design; I would suggest to indicate more clearly that the "reference" experiments were carried out over a discrete set of globally uniform "b" values (how many?), and to distinguish this more clearly from the LHS experiments for the regional variation

Eqn. 2 and Table 1: The connection between beta_0 and beta_k of Eqn 2 and Table 1 is not clear to me: are beta in the table beta_k of equation 2? Is beta_0 constant?

p 5, line 11: "we fit linear regression models" - I suggest to refer here again to Eqn 2.

p 5, line 28-29: "However, the relative sensitivity ranked across regions remains similar, as shown by expressing b_k as a percentage (Table 1)." - relative to what?

Table 1: Please explain clearly what is shown in this Table: are beta the beta_k of Eqn 2? What does beta(%) mean - normalised by area? Are the two rightmost columns for the constant export experiments?

p 6 line 4 "positive"

p 6 second paragraph: is there a difference between "export production" and "export productivity"?

p 6, line 20 "is normalised" - by what?

p 7 line 3: "sensitivity"

p 11, line 5: "$\kappa$"

p 11, line 20: "function"

p 19, caption: "relects"?
* * *

---

## Referee Comment (RC3) · Anonymous Referee #3 · 28 Jan 2019

This paper presents a novel approach to answering an important open question. The precise pattern of spatial variability in how organic carbon is remineralised is still debated, so the authors approach the issue instead from the perspective of "where are the most sensitive regions?", with sensitivity defined as changes in atmospheric CO2 through the proxy of preformed phosphate. This is a nice piece of work and I'm happy to recommend publication. I make some suggestions below, largely relating to clarity on technical points.

- While I understand why the two model scenarios (restoring or fixed export) are presented as end-members, the fixed export run nevertheless takes its export from a

[Figure]

restoring run. It is true that output from the run giving the closest fit to observations is used as baseline but it should still be acknowledged that the 'end-members' are far from independent models.

- The description of tracking preformed phosphate needs more detail. The decomposition described in Appendix B gets phosphate away from surface only – it still needs to be tracked in the interior. How is this done?

- The authors should show the scatter plot of predicted vs observed values for the relationship described in page 5 lines 12-13 as it is fundamental to the manuscript. It should show predicted and observed changes in PO4 as this is the predicted field.

- Fig 6 and section 3.2 – there is a sound argument for geometric mean so just show geometric mean and give the argument in the methods. It is not necessary to show arithmetic mean results in Fig 6a

- As a more informative second panel for Fig 6 show the same as current 6b but with regression taken out to show variability due to regional variability more clearly. The authors should also acknowledge in the text that the random sampling leads to under-sampling of highest and lowest global b values.

- How independent in structure are the 3 models used for the PO4 vs pCO2 relation?

- Fig A1 should be in the main body of the paper

- Consistency needed in terminology: in Subantarctic (text) and subpolar (fig)

- Remineralisation depth is defined (page 2, lines 8-9) assuming exponential profile (decrease by 63%) - but models use Martin curve

- Does the misfit function used to carry out the comparison to WOA (page 4, lines 22-23) take volume into account?

- Explain the maximin Matlab option for hypercube sampling in Matlab (page 4, line 31)

- page 4 line 25: not sure that "reference" is appropriate

- Fig 3 caption needs rewording. All values are positive.

- The authors' definition of the Subantarctic boundary makes it a little difficult to compare results to Kwon's paper where the Southern Ocean was defined as south of 40S. Given that the Kwon paper provides such strong motivation for this manuscript this deserves comment.

- Page 6, lines 4-6: It should be explicitly acknowledged that there is a rather weak relationship between export and sensitivity for the restoring runs (Fig 4b)

- Use notation that distinguishes regional and global means of PO4_pre

- Both constant export and nutrient restoring should be shown in Fig 5.

- Page 7, Line 3: "sensitivity"

- Page 8, lines 28-30: "As such, the global mean change in potential future and past changes in remineralisation depth may be larger than the uncertainty associated with spatial variability." B changes discussed less than current observed range" The changes being discussed here are substantially smaller than the current range of observed values. Even if, as this paper argues, the global ocean may not be overly sensitive to spatial variation in b, it is worth noting that the current uncertainty in a global value of b still has very large uncertainty partly because of the confounding effect of under-sampled spatial variability.

- Page 10, line 21: Which sea ice field is used?

- Page 11, line 5: 1-v not 1-kappa

- Appendix A: state that the bottom of the second grid box in the vertical is at 120m (presumably)?

---

## Referee Comment (RC4) · Anonymous Referee #4 · 30 Jan 2019

The submission by Wilson et al explores the sensitivity of atmospheric CO2 to regional variability in particulate organic matter remineralization depth in the ocean. In doing so, it ties together two sets of important previous findings: those showing that remineralization depth is a key lever on global climate, and those showing that remineralization depth varies significantly between regions. The manuscript uses some interesting methods, such as breaking interior nutrient distributions into components that circulated from different surface regions. It also reaches some interesting conclusions, including the important role of the Subantarctic region, and the fact that regional changes in remineralization depth might be closely approximated by a corresponding uniform change in the global flux profile. However, I have two major concerns that

mean I cannot support publication of the paper in its current form.

First, I am concerned that biases in the circulation model might skew the results and have not been properly acknowledged. Can the authors state whether they are using a new version of the MITgcm 2.8degree circulation model, or the same one that has been used since the early OCMIP era? Previous studies (Dutay et al. 2002; Doney et al. 2004) have identified some significant shortcomings of this circulation model that might impact the relative importance of different regions in the current study. Not least, the model does not produce deep water along the Antarctic coastline as it should, and instead produces deep water at around 50S. This would shift deep water formation from the Antarctic to the Subantarctic regions defined in the current study, and give the Subantarctic region unrealistic leverage over interior nutrient distributions. It could be the case that the circulation has been reformulated since those studies and this bias corrected. If that's the case it is important for the authors to demonstrate this, to reassure readers like myself who have reservations about that model. The simplest way to show this would be to calculate ideal age in their model, and plot a meridional cross section through the Pacific. They should be able to show a tongue of young water subducting right along the Antarctic coastline and spreading northwards along the seafloor (not a tongue of young water penetrating the deep ocean at 50S). If they are indeed using the old, biased circulation model, this should be acknowledged in the text where the significance of the Subantarctic region is discussed. Either way, a figure like the one I suggested should be included as a supplementary figure either to demonstrate that the circulation model is robust, or to make readers aware of potential biases introduced by the Southern Ocean wartermass structure.

Second, I am confused as to why a paper focused on sensitivity of atmospheric CO2 does not use a model that resolves the carbon cycle. Instead, they model only the phosphorous cycle and relate it to carbon cycle changes using a relationship derived from prior modeling studies. The authors state that this is to avoid the computational expense of simulating the carbon cycle. But that would only require the addition of two

tracers – DIC and Alk, and a single value for a well-mixed atmospheric CO2 concentration. This should therefore only double the computation time, and given that transport matrix method is being used (where efficient Crank-Nicolson time-stepping methods can be applied), this does not seem preclusive. And even then, they needn't include the carbon cycle in all 200 of their simulations, only enough to redefine the statistical PO4 vs. CO2 relationship from their own model. This would at least keep their study self-consistent, rather than relying on previous results from different models. If the authors are not able to do this in the current study (which would be preferable), they should again acknowledge more clearly the caveats of their chosen method. In Fig. A1, it is obvious that different models yield different relationships between these properties. Fitting just the Marinov et al. results would lead to a much shallower relationship, but those results are not strongly weighted because they contribute fewer data points than others. It would seem more reasonable to fit the relationship for each previous study separately, and propagate that uncertainty into their CO2 estimates.

Minor points

Figure 3. I think a bar plot would be better suited to show this data. This is a key result of the paper, and a quantitative comparison between regions and production methods would be simpler in a bar format.

Figure 5. This figure is a little overcomplicated, as evidenced by the fact that a fair amount of the text (not just the caption) is devoted to explaining what it means. How about showing these results as a color matrix instead? Region in which b is varied down the rows, region in which we are looking at the preformed PO4 along the columns (or vice versa), color shows the regression coefficient. I know this is contrary to my previous comment about bars being more precise, but I think precision is less important here than the need to reduce complexity. The color matrix would allow the reader to pick out "bright" rows or columns as indicative of important regions.

Section 3.2 and Figure 6. I'm not sure why the results in Figure 6a are shown? The

authors acknowledge that simply averaging the b values is not the correct way to quantify the global-mean remineralization profile, and then attempt to correct for it in panel b. But why show an obviously incorrect result in the first place? It seems like the correct way to define a "global mean b-value" would be to construct a global-mean (area-weighted) organic matter flux profile, and then fit the Martin relationship to that.

---

## Author Comment (AC1) · 10 Apr 2019

We thank Reviewer #4 for their constructive comments. We have listed their comments in bold below and our responses in normal formatting.

*Reviewer #4*

**First, I am concerned that biases in the circulation model might skew the results and have not been properly acknowledged. Can the authors state whether they are using a new version of the MITgcm 2.8degree circulation model, or the same one that has been used since the early OCMIP era? Previous studies (Dutay et al. 2002; Doney et al. 2004) have identified some significant shortcomings of this circulation model that might impact the relative importance of different regions in the current study. Not least, the model does not produce deep water along the Antarctic coastline as it should, and instead produces deep water at around 50S. This would shift deep water formation from the Antarctic to the Subantarctic regions defined in the current study, and give the Subantarctic region unrealistic leverage over interior nutrient distributions. It could be the case that the circulation has been reformulated since those studies and this bias corrected. If that's the case it is important for the authors to demonstrate this, to reassure readers like myself who have reservations about that model. The simplest way to show this would be to calculate ideal age in their model, and plot a meridional cross section through the Pacific. They should be able to show a tongue of young water subducting right along the Antarctic coastline and spreading northwards along the seafloor (not a tongue of young water penetrating the deep ocean at 50S). If they are indeed using the old, biased circulation model, this should be acknowledged in the text where the significance of the Subantarctic region is discussed. Either way, a figure like the one I suggested should be included as a supplementary figure either to demonstrate that the circulation model is robust, or to make readers aware of potential biases introduced by the Southern Ocean watermass structure.**

Thank you for highlighting this important caveat. We have plotted the meridional cross section (Figure 1) and the model does subduct water around 50°S. In response to this, and to comments from other reviewers, we have included this figure, a comparison of where the densest surface waters are versus observations, and a water-mass analysis in the supplementary material. We have also updated the Discussion in the manuscript to make clear that this is an important caveat. We have kept the circulation model for a number of reasons: 1) the Subantarctic regions do not dominate the sensitivity at a global level, 2) the circulation model is likely to over-estimate the sensitivity of $CO_2$ to remineralisation in the Subantarctic regions, 3) it is widely used for modelling biogeochemistry.

[Figure]

Figure 1. Meridional cross section of ideal age in the Pacific (224°W).

**Second, I am confused as to why a paper focused on sensitivity of atmospheric CO2 does not use a model that resolves the carbon cycle. Instead, they model only the phosphorous cycle and relate it to carbon cycle changes using a relationship derived from prior modeling studies. The authors state that this is to avoid the computational expense of simulating the carbon cycle. But that would only require the addition of two tracers – DIC and Alk, and a single value for a well-mixed atmospheric CO2 concentration. This should therefore only double the computation time, and given that transport matrix method is being used (where efficient Crank-Nicolson time-stepping methods can be applied), this does not seem preclusive. And even then, they needn't include the carbon cycle in all 200 of their simulations, only enough to redefine the statistical PO4 vs. CO2 relationship from their own model. This would at least keep their study self-consistent, rather than relying on previous results from different models. If the authors are not able to do this in the current study (which would be preferable), they should again acknowledge more clearly the caveats of their chosen method. In Fig. A1, it is obvious that different models yield different relationships between these properties. Fitting just the Marinov et al. results would lead to a much shallower relationship, but those results are not strongly weighted because they contribute fewer data points than others. It would seem more reasonable to fit the relationship for each previous study separately, and propagate that uncertainty into their CO2 estimates**

In response to this comment, and to comments from other reviewers, we have added the carbon cycle to the model. We have redefined the preformed $PO_4$ and $CO_2$ relationship and have used this to calculate the change in $CO_2$ for the Latin hypercube ensemble.

**Figure 3. I think a bar plot would be better suited to show this data. This is a key result of the paper, and a quantitative comparison between regions and production methods would be simpler in a bar format.**

Thank you for this suggestion. We have added additional panels to the figure showing the sensitivity estimates in a bar format (Figure 2).

[Figure]

Figure 2. Regional sensitivity ($\Delta CO_2$ / $\Delta b$) of atmospheric $CO_2$ (ppm) to changes in Martin curves for the constant-export scheme (panels a & b) and restoring-uptake scheme (panels c & d). Atmospheric $CO_2$ is inferred from modelled preformed $PO_4$ using empirical relationships.

**Figure 5. This figure is a little overcomplicated, as evidenced by the fact that a fair amount of the text (not just the caption) is devoted to explaining what it means. How about showing these results as a color matrix instead? Region in which b is varied down the rows, region in which we are looking at the preformed PO4 along the columns (or vice versa), color shows the regression coefficient. I know this is contrary to my previous comment about bars being more precise, but I think precision is less important here than the need to reduce complexity. The color matrix would allow the reader to pick out "bright" rows or columns as indicative of important regions.**

Thank you for this useful suggestion. We have reformatted the figure as a matrix (Figure 3). This does highlight spatial patterns more clearly and allows both constant-export and nutrient-restoring results to be shown.

[Figure]

Figure 4. Sensitivity of steady-state normalised mean preformed [PO₄] exported from each region. The preformed [PO₄] from each region is expressed as a function of b using linear regression. Preformed [PO₄] is normalised to the range of values in the ensemble to account for large differences in preformed [PO₄] between regions. The regression coefficients are arranged such that each row shows the impact of changing b in that region on preformed [PO₄] across other regions. Results from the constant-export and nutrient-restoring schemes are shown in the top and bottom panels respectively.

**Section 3.2 and Figure 6. I'm not sure why the results in Figure 6a are shown? The authors acknowledge that simply averaging the b values is not the correct way to quantify the global-mean remineralization profile, and then attempt to correct for it in panel b. But why show an obviously incorrect result in the first place? It seems like the correct way to define a "global mean b-value" would be to construct a global-mean (area-weighted) organic matter flux profile, and then fit the Martin relationship to that.**

We have moved panel a to the discussion of averaging in the supplementary material. We have kept the averaging approach as this has been used previously, e.g., Henson et al., (2012), and so provides useful context.

---

## Author Comment (AC2) · 10 Apr 2019

We thank Reviewer #3 for their constructive comments. We have listed their comments in bold below and our responses in normal formatting.

*Reviewer #3*

**While I understand why the two model scenarios (restoring or fixed export) are presented as end-members, the fixed export run nevertheless takes its export from a restoring run. It is true that output from the run giving the closest fit to observations is used as baseline but it should still be acknowledged that the 'end-members' are far from independent models.**

We have amended the text as follows: "These two schemes represent two end-member scenarios, strictly within the context of this model, where organic matter production either depends entirely on macronutrient concentrations…"

**The description of tracking preformed phosphate needs more detail. The decomposition described in Appendix B gets phosphate away from surface only – it still needs to be tracked in the interior. How is this done?**

We have changed equation B1 in Appendix B to make the operation clearer:

$$PO_4^{pre} = (A_i^I A_e^I - I)^{-1}((A_i^I B_e + B_i)PO_4)$$

**The authors should show the scatter plot of predicted vs observed values for the relationship described in page 5 lines 12-13 as it is fundamental to the manuscript. It should show predicted and observed changes in PO4 as this is the predicted field.**

We have added this figure to the manuscript, (see Figure 1 here).

[Figure]

Figure 1. Residuals for the linear regressions that estimate sensitivity of $CO_2$ to spatially varying Martin curves for (a) constant-export and (b) restoring-uptake schemes.

**Fig 6 and section 3.2 – there is a sound argument for geometric mean so just show geometric mean and give the argument in the methods. It is not necessary to show arithmetic mean results in Fig 6a**

We have moved this figure panel to the discussion of calculating the geometric mean in the supplementary material.

**As a more informative second panel for Fig 6 show the same as current 6b but with regression taken out to show variability due to regional variability more clearly. The authors should also acknowledge in the text that the random sampling leads to undersampling of highest and lowest global b values.**

Thank you for this suggestion. Because the global mean of responses track the globally-uniform responses closely and we could not find any evidence that the variability was associated with changes in b in any specific region, this additional plot did not provide much additional information so we have kept the original panel.

We have added the following text: "Note that b in each region is varied within the full parameter range but that because Latin hypercube sampling varies all parameters across their parameter range simultaneously the global mean does not reach the highest and lowest global b values."

**How independent in structure are the 3 models used for the PO4 vs pCO2 relation?**

In response to other reviewer comments we have replaced the statistical relationship between preformed $PO_4$ and $CO_2$ with one calculated specifically for this model using a carbon cycle.

**Fig A1 should be in the main body of the paper**

We have moved the equivalent plot to the Methods section of the manuscript.

**Consistency needed in terminology: in Subantarctic (text) and subpolar (fig)**

We have changed any use of 'subpolar' to 'Subantarctic' throughout.

**Remineralisation depth is defined (page 2, lines 8-9) assuming exponential profile (decrease by 63%) but models use Martin curve**

The reviewer is correct that the definition assumes exponential decay whereas the Martin curve is a power-law. This was used previously by Kwon *et al.,* (2009) who used Martin curves but also expressed them as e-folding depths. Our purpose was to introduce the term 'remineralisation depth' as this allows for more clear and concise discussion of changes in Martin curves as the terms 'shallower' or 'deeper' can be used rather than changes in the dimensionless exponent b.

We have changed the text to better reflect this comment:

"In this paper we use the term 'remineralisation depth', defined as a depth at which a defined % of POC has been remineralised. Previously, this has been defined as an *e*-folding depth: the depth at which ~63% of POC has been remineralised (Kwon et al., 2009) (although note the Martin curve is not exponential)."

**Does the misfit function used to carry out the comparison to WOA (page 4, lines 22-23) take volume into account?**

Yes. We have amended the text to state that it is volume-weighted.

**Explain the maximin Matlab option for hypercube sampling in Matlab (page 4, line 31)**

We have added the following text to clarify:

"…with 'maximin' sampling (an additional constraint that helps reduce clustering of samples, by maximising the minimum distance between points, in order to give a well-spread distribution of points across the parameter space)."

**page 4 line 25: not sure that "reference" is appropriate**

We have updated the experiment description with headings to separate the description of the control run, global and regional sensitivity runs. "Reference" has been removed from the text.

**Fig 3 caption needs rewording. All values are positive.**

The caption has been reworded to: "The sensitivity value reflects the increase in $CO_2$ (preformed $PO_4$) for an increase in b (shallower remineralisation)."

**The authors' definition of the Subantarctic boundary makes it a little difficult to compare results to Kwon's paper where the Southern Ocean was defined as south of 40S. Given that the Kwon paper provides such strong motivation for this manuscript this deserves comment.**

We had added an additional row to Table 1 in the manuscript describing metrics for the Southern Ocean as defined as >38°S for comparison with Kwon et al., (2009). We have also noted this comparison in the Discussion.

**Page 6, lines 4-6: It should be explicitly acknowledged that there is a rather weak relationship between export and sensitivity for the restoring runs (Fig 4b)**

We have added correlation coefficients to help demonstrate the weaker relationship between export and sensitivity. The following text has been added:

"Similarly, we find a general positive correlation between sensitivity and regional export production (r=0.79, p<0.01 for constant export, r=0.47, p=0.07 for restoring uptake), as measured by the mean annual average export production across the 200 ensemble runs (Fig 4). The correlation is much weaker with nutrient restoring uptake compared to the constant-export production."

**Use notation that distinguishes regional and global means of PO4_pre**

We have updated the text with notation to distinguish between regional and global means of preformed $PO_4$.

**Both constant export and nutrient restoring should be shown in Fig 5.**

In response to other comments from reviewers, we have updated the format of this figure and have added panels for both constant export and nutrient restoring.

**Page 7, Line 3: "sensitivity"**

Fixed.

**Page 8, lines 28-30: "As such, the global mean change in potential future and past changes in remineralisation depth may be larger than the uncertainty associated with spatial variability." B changes discussed less than current observed range" The changes being discussed here are substantially smaller than the current range of observed values. Even if, as this paper argues, the global ocean may not be overly sensitive to spatial variation in b, it is worth noting that the current uncertainty in a global value of b still has very large uncertainty partly because of the confounding effect of under-sampled spatial variability**

We have added the following text: "However, we note that the modern global mean b is subject to uncertainty associated with under-sampled spatial variability."

**Page 10, line 21: Which sea ice field is used?**

The following text has been added:

"…scaled the fraction of seaice present (Fice, as monthly average fields from the original circulation model)."

**Page 11, line 5: 1-v not 1-kappa**

Fixed.

**Appendix A: state that the bottom of the second grid box in the vertical is at 120m (presumably)?**

Done.

---

## Author Comment (AC3) · 10 Apr 2019

We thank Reviewer #2 for their constructive comments. We have listed their comments in bold below and our responses in normal formatting.

*Reviewer #2*

**My main concern is that the results are likely tied to the circulation model applied. As shown by Duteil et al. (2013; Biogeosciences, 10, 7723–7738, doi:10.5194/bg-10-7723-2013) the transport matrices from the MITgcm seem to suffer from far too large outcrop areas of dense waters in the Southern Ocean (their Fig. 2), indicating that the model circulation does not represent the real ocean in that region very well. Also, because of the very coarse resolution, this model might not represent the physical dynamics in the eastern equatorial Pacific very well. However, in the present study these two regions - the subantarctic regions and equatorial upwelling - have a large influence on CO2 (Fig. 3 and 5). Thus, whereas this study provides important and interesting information for other global model studies that apply similar circulations (as noted in Discussion and Conclusions), I think that a few sentences on this are necessary to caution readers not familiar with the advantages and disadvantages of global circulation models. (To illustrate or investigate this point further, one could, e.g., look at the density distribution or mixed layer depths of the model.)**

We have added a plot of density outcrops from the annual mean model output and World Ocean Atlas 13 climatological observations (see Figure 1), a comparison of the volume of water ventilated from each region in the model with the data-constrained estimates from Khatiwala *et al.,* (2012) (Table 1), and a plot of ideal mean age (Figure 2) to the Supplementary Material to demonstrate this caveat.

As the reviewer highlights, the modelled Subantarctic regions are a larger source of water for the ocean interior than observed (Figure 1, Table 1). Additionally, the equatorial regions contribute a much smaller volumetric fraction than observed (Table 1). An alternative approach could be to use the data-constrained ECCO circulation but this comes with a much higher computational cost due to higher resolution and higher number of non-zeros in the sparse matrices, limiting the feasibility of the sensitivity analysis. The MITgcm circulation, as noted by the reviewer, has been widely applied. Therefore, we have kept the MITgcm circulation and have added a substantial discussion in the manuscript referring to the new supplementary figures that discusses the circulation as a caveat to the findings:

"Our results are dependent on the use of transport matrices derived from one global circulation model. Whilst this model has been widely applied to study biogeochemistry previously, it is subject to a number of caveats. The ocean model predicts significantly larger outcrops of dense water in the Southern Ocean compared to observations (see Figure S4; Duteil et al., 2013) leading to deep-water formation occurring at latitudes around 50S (Figure S5). The volumetric fraction of water in the ocean interior derived from the Subantarctic is also higher (26%) compared with data-constrained estimates (18%: Khatiwala et al., 2012). As such, the sensitivity estimates for the Subantarctic may be over-estimated. This is also consistent with the higher sensitivity compared to the basin-scale analysis of Kwon et al., (2009) who found that the Southern Ocean (>40S contributed 22% of the global $CO_2$ sensitivity, compared with 36% in this study (>38S, Table 1). However, our results have key similarities, including absolute and relative magnitudes of regional preformed $PO_4$ export, to other studies using alternative steady-state circulation states (DeVries et al, 2012; Pasquier and Holzer 2016). As such, our results should be broadly reproducible with other models."

[Figure]

Figure 1: Regions where density is greater than 1027.5 kg m$^{-3}$ calculated using the Gibbs SeaWater toolbox (McDougall & Barker 2011) with annual-mean temperature and salinity from (a) World Ocean Atlas 18 and (b) MITgcm output.

Figure 2. Meridional cross section of ideal age in the Pacific (224°W).

[Figure]

Table 1. Global ocean volumetric fraction (%) for different source regions from a data-constrained estimate (Khatiwala *et al.,* 2012) and from this study.

| Region | Khatiwala2012 (%) | This Study (%) |
|---|---|---|
| Antarctic | 39 | 28.7 |
| Subantarctic | 18 | 26.1 |
| North Atlantic | 26 | 35 |
| Tropical | 4.5 | 0.86 |
| Subtropics | 8.1 | 4.5 |
| NPacific | 4 | 4.5 |

**There seems to be a strong sensitivity of CO2 to changes in b in the constant-export scheme (Fig 3), and also a clear relationship to export (Fig 4a). In contrast, normalized (by what?) preformed phosphate seems to be more sensitive in the nutrient restoring scenario (Fig 5 vs Fig S3), and no relationship seems to exist between CO2 sensitivity and export (Fig 4b). I think these contrasting patterns for both model types deserve a bit more discussion. Perhaps some section plots of, e.g., density across the Pacific and Atlantic (see above) could aid the disussion about the effects of circulation vs. export type ("biogeochemistry"). If the circulation model is anywhere near the real world, some insight regarding the "connectivity" of different regions might perhaps be gained from the data-constrained analysis of water fractions presented by Khatiwala et al. (2012; Earth and Planetary Science Letters, 325–326, 116–125)**

In response to this and other reviewer comments, we have replotted Figure 5 in a format which is hopefully more accessible, and that allows for the inclusion of panels for both the fixed and restoring export ensembles (see Figure 3 here). The new plot highlights that the sensitivity estimates are broadly similar across both the nutrient-restoring and constant-export schemes but that preformed $PO_4$ appears more sensitive to local changes in b (boxes on the diagonal) in the nutrient-restoring scheme. We have added text in the Results to note this difference. We have also included the following equation in the manuscript text to describe the normalisation:

$$\overline{P_{pre}^{region}} = \frac{\overline{P_{pre}^{region}} - \min(\overline{P_{pre}^{region}})}{\max\left(\overline{P_{pre}^{region}}\right) - \min(\overline{P_{pre}^{region}})}$$

We have addressed the comments on circulation in the response to the previous comment.

In terms of the relationship between sensitivity and export production, the distribution of regions within the nutrient-restoring panel is very similar to the constant-export panel despite a much weaker relationship. We have added text to the Results to demonstrate the weaker relationship between export production and sensitivity for the nutrient-restoring scheme:

"Similarly, we find a general positive correlation between sensitivity and regional export production (r=0.79, p<0.01 for constant export, r=0.47, p=0.07 for restoring uptake), as measured by the mean annual average export production across the 200 ensemble runs (Fig 4). The correlation is much weaker with nutrient restoring uptake compared to the constant-export production."

[Figure]

Figure 3. Sensitivity of steady-state normalised mean preformed [$PO_4$] exported from each region. The preformed [$PO_4$] from each region is expressed as a function of b using linear regression. Preformed [$PO_4$] is normalised to the range of values from each region within the ensemble to account for large differences in preformed [$PO_4$] between regions. The regression coefficients are arranged such that each row shows the impact of changing b in that region on preformed [$PO_4$] across other regions. Results from the constant-export and nutrient-restoring schemes are shown in the top and bottom panels respectively.

**p 3, line 15: "MITgcm" sounds like technical slang to me - is there a better word for it?**

The text has been changed to: "*MIT general ocean circulation model (MITgcm)*"

**Section 2.3: At first, I had difficulties understanding the experimental design; I would suggest to indicate more clearly that the "reference" experiments were carried out over a discrete set of**

**globally uniform "b" values (how many?), and to distinguish this more clearly from the LHS experiments for the regional variation**

We have updated the text with headings to separate the description of the control run, global and regional sensitivity runs. We have also clarified the number of globally uniform b values tested.

**Eqn. 2 and Table 1: The connection between beta_0 and beta_k of Eqn 2 and Table 1 is not clear to me: are beta in the table beta_k of equation 2? Is beta_0 constant?**

We have added the subscripts to the betas in Table 1 and have added a reference to eqn. 2 in the Table caption.

**p 5, line 11: "we fit linear regression models" - I suggest to refer here again to Eqn 2.**

Done.

**p 5, line 28-29: "However, the relative sensitivity ranked across regions remains similar, as shown by expressing b_k as a percentage (Table 1)." - relative to what?**

This has been reworded to "…as shown by expressing each $\beta_k$ as a percentage of $\sum_k \beta_k$ (Table 1)."

**Table 1: Please explain clearly what is shown in this Table: are beta the beta_k of Eqn 2? What does beta(%) mean - normalised by area? Are the two rightmost columns for the constant export experiments?**

We have added the subscripts to the betas in Table 1 and have added a reference to eqn. 2 in the Table caption. We have also added annotation and text to the caption to explicitly state that the beta(%) is relative to the sum of the regression coefficients.

**p 6 line 4 "positive"**

Fixed.

**p 6 second paragraph: is there a difference between "export production" and "export productivity"?**

Fixed. Export production is now used throughout.

**p 6, line 20 "is normalised" - by what?**

The text has been updated to explicitly describe the normalising (see also equation above)

**p 7 line 3: "sensitivity"**

Fixed.

**p 11, line 5: "$\kappa$"**

Fixed.

**p 11, line 20: "function"**

Fixed.

**p 19, caption: "relects"?**

Fixed to "reflects".

---

## Author Comment (AC4) · 10 Apr 2019

We thank Reviewer #1 for their constructive comments. We have listed their comments in bold below and our responses in normal formatting.

*Reviewer #1*

**Firstly, the model only infers the atmospheric CO2 response based on the published relationship between the oceanic inventory change in preformed PO4 and the atmospheric CO2 change. The near-linear relationship between the two is only valid if we assume no solubility and air-sea CO2 disequilibrium effects. I acknowledge that the effects might be implicitly included in the empirical relationship extracted from a few previous global studies. However, I am not sure if the same relationship can be applied to the regional perturbation study. For example, any perturbations in the Southern Ocean (e.g., the ACC band where air-sea CO2 disequilibrium is large due to the short surface residence time of upwelled waters) might not lead to the atmospheric CO2 response proportional to the preformed PO4 response. This point can be especially worrisome because the most sensitive regions turn out to be the ACC band in the study.**

Thank you for highlighting this. We have replaced the published preformed $PO_4$ / $CO_2$ relationship with one calculated explicitly for this model using an online carbon cycle.

We have tested the impact of $CO_2$ disequilibrium on the sensitivity results by running an ensemble of experiments where the Martin curve in each region is perturbed individually whilst all others are kept at the control value. The runs have a carbon cycle and so atmospheric $CO_2$ is predicted as a response to the changing biogeochemistry. The preformed $PO_4$ and $CO_2$ from each regional perturbation are plotted over the values from global perturbation experiments (Figure 1). There are minor deviations from the global preformed $PO_4$ / $CO_2$ relationship suggesting that disequilibrium effects may be present, but these are relatively minor. The range of $CO_2$ values predicted by the individual perturbation experiments closely matches the sensitivity patterns from the Latin Hypercube ensemble (Figure 2) further suggesting disequilibrium effects have a minor impact on the results.

We have added these results to the supplementary material that are referenced from a brief discussion of disequilibrium effects in the methods section of the manuscript. We highlight in the discussion that disequilibrium effects may be considered in future analyses.

[Figure]

Figure 1. Comparison of $CO_2$ versus preformed $[PO_4]$ relationships when the Martin curve is varied as a globally uniform value (black line) from -0.4 to -1.6, and when regions are perturbed individually within the same range (grey). Panels (a) and (b) shows results for the constant-export ensemble with the region of interest expanded in (b). Panels (c) and (d) shows results for the nutrient-restoring ensemble with the region of interest expanded in (d).

[Figure]

Figure 2. Comparison of $CO_2$ sensitivity estimates from two methods. Black bars are the Latin hypercube and regression-based sensitivity estimates derived with the statistical relationship between preformed $PO_4$ and $CO_2$. Grey bars are the difference between $CO_2$ when each region is perturbed individually from b=0.2 and b=1.6 and atmospheric $CO_2$ is calculated explicitly in the model.

**Secondly, the sensitivity is estimated using the multiparameter linear regression method applied to the two sets of 200-member ensemble experiments where 15 regional exponents are perturbed simultaneously. Although the method seems sound, I wonder why the sensitivity should be quantified in this way? Are there any merits? Would the sensitivity be the same or different if the authors perturbed the exponent in a region at a time, requiring a total of 15 perturbation experiments for one model scheme? The individual perturbation experiments seem a simpler and cleaner way to quantify the atmospheric CO2 response to the perturbation and its relations with the export production in each domain.**

The reviewer is correct that an alternative approach could be to perturb the remineralisation depth in each region individually (although this would entail at least 4 experiments per region to reasonably characterise the response across the parameter range). The advantage of the Latin hypercube

sampling approach is that the resulting sensitivity combines the direct influence of changing remineralisation in one region of interest plus the joint influence of remineralisation changes in other regions, e.g., Pianosi *et al.,* (2016). Given the uncertainty in the observed spatial distribution of remineralisation and the driving mechanisms, we feel it is important to account for the full range of simultaneous changes in remineralisation depths and keep this as the main result.

As per the previous response, we have run the individual perturbation experiments with an explicit carbon cycle. The Martin curve in each region is perturbed to -0.4, -0.7, -1.3 and -1.6 whilst other regions are maintained at the control value of -1.0. Figure 2 shows $\Delta CO_2$ from the -0.4 and -1.6 experiments for each region. The sensitivity patterns match the results from the Latin hypercube ensemble closely. There are differences in magnitude between the two estimates likely due to the fact that there are minor interactions between regions. However, the patterns between regions are preserved. We have added these results in the supplementary as this will provide important context for the key results and strengthen the analysis and interpretation.

**Thirdly, the major novel finding is that the highest sensitivity in atmospheric CO2 is to the change in remineralization depth in Subantarctic regions due to high export production and the high connectivity to deep water formation regions. I see the reasoning behind it: The export production should be high because the export will determine how much regenerated PO4 can be affected by the perturbation. The connectivity to deep water formation regions is important because the deep water formation is the main pathway of preformed PO4 to the ocean's interior and the inventory of preformed PO4. However, I am not fully convinced by the authors' finding. Both "nutrient restoring" model and "constant export" model show that the sensitivity of atmospheric CO2 to the remineralization depth change is also high in the "NTemp-PAC" domain (Fig. 3). Yet, the subtropical North Pacific is not a region with high export production nor close to any deep water formation regions. How can it be explained?**

Thank you for highlighting this. It is notable that the NTemp-PAC region in general, for this model, does fall along the general trend between export production and sensitivity. In comparison the STemp-PAC region has a much lower sensitivity for a similar export production. This could be related to the age of water masses in the Pacific whereby deeper remineralisation in the STemp-PAC region sequesters organic matter in much younger waters, that will return to the surface ocean faster, thus reducing the sensitivity. We have already noted the range of sensitivity for similar export production in the manuscript, so we have added additional text to discuss this. Overall, because the spatial variability is not overly significant relative to the global uniform variability and because export production is a strong predictor of sensitivity, we have not explored this further.

**Similarly, why are the deep water formation regions (i.e., the model NADW and AABW formation regions) not the sensitive regions?**

We have added the contribution of preformed $PO_4$ derived from each region for the control run in Table 1 to help highlight the sensitivity of the deep water formation regions. Kwon et al., (2009) demonstrated that largest changes in preformed $PO_4$ when changing b globally were associated with deep water formation regions. The updated version of Figure 5 (Figure 4 here), shows that this is in response to changes in b from globally distributed regions (compare colours across the column for SO and NN-Atl regions).

**It may be related to the third point. But I don't understand Figure 5. The sensitivity is normalized to what? What do the authors mean by "mean preformed PO4 in a region"? Is it the surface PO4**

**averaged over each region or the total preformed PO4 subducted from each region divided by the volume of water subducted from the same region?**

The "mean preformed PO$_4$ in a region" is calculated in a similar way to the global preformed PO$_4$ concentration: the annual mean surface [PO$_4$] at the end of a simulation is set as a boundary condition and the transport matrix used to calculate the interior distribution of preformed PO$_4$. In this case, the annual mean surface [PO$_4$] is set in the region of interest only and the remaining surface has concentrations of zero. The global mean concentration is then calculated from the interior distribution.

Because the absolute magnitude of regional mean preformed PO$_4$ varies by an order of magnitude across all regions, it is necessary to normalise the concentrations to allow comparison of regression coefficients using the following relationship:

$$\overline{P_{pre}^{region}} = \frac{\overline{P_{pre}^{region}} - \min(\overline{P_{pre}^{region}})}{\max\left(\overline{P_{pre}^{region}}\right) - \min(\overline{P_{pre}^{region}})}$$

In response to this point and to comments from other reviewers, we have revised the format of the figure (see Figure 4 here). We have also updated the text with the description above and the equation to clarify the method of obtaining preformed PO$_4$ and normalisation.

[Figure]

Figure 4. Sensitivity of steady-state normalised mean preformed [PO$_4$] exported from each region. The preformed [PO$_4$] from each region is expressed as a function of b using linear regression. Preformed [PO$_4$] is normalised to the range of values in the ensemble to account for large differences in preformed [PO$_4$] between regions. The regression coefficients are arranged such that each row shows the impact of changing b in that region on preformed [PO$_4$] across other regions. Results from the constant-export and nutrient-restoring schemes are shown in the top and bottom panels respectively.

---

## Referee Report (RR1)

Review of "Sensitivity of atmospheric $CO_2$ to regional variability in particulate organic matter remineralization depths" by Wilson et al.

Overall, the manuscript has become clearer and the authors have addressed most of my comments. However, some parts in Article and especially in Suppl. Material remain still unclear, as illustrated below. It is also confusing that the authors did not distinguish the figures in Suppl. Material (usually denoted as Figure S1, S2, etc.) from the figures in the main Article.

I appreciate the additional sensitivity experiments where atmospheric CO2 is explicitly simulated, instead of being inferred from preformed PO4 changes in the model. In the manuscript, the authors stated that atmospheric CO2 was explicitly simulated in the "global" sensitivity experiment (page 4, line 29-page 5, line 11) and that the relationship between preformed PO4 and atmospheric CO2 was derived from Figure A1. In Suppl. Material and the response letter, the authors also stated that atmospheric CO2 was also explicitly simulated in the regional sensitivity experiments and added Figure 1. From which figure do the authors derive the relationship between preformed PO4 and atmospheric CO2 when estimating the regional sensitivity of atmospheric CO2 to change in b from the Latin Hypercube ensemble?

Likewise, it is difficult to understand the newly added Figures 1 though 3 in Suppl. Material. For example, what is the difference between Figure 1a and 1b, and between Figure 1c and 1d? What do the authors mean by "region of interest expanded"? Why do the relationships between preformed PO4 and atmospheric CO2 differ between Figure 1a (linear) and 1b (non-linear)? Which relationship is Figure 3 based on? In Figure 2, the studentized residuals seem large, on the contrary to the residuals shown in Figure 1. Why? In Figure 3, the magnitudes of CO2 sensitivity look different between the black bars and the gray bars (not only for the absolute values but also for the relative values across the regions), although the authors claim that there is a good agreement. I feel that the authors need to discuss in what aspects the two (black vs. gray) do not agree and in what aspects the two agree in more details.

In Figure 3 and Table 1 of the main article, it would be useful if the authors can add the error bars for the regional CO2 sensitivity (e.g., the 95% confidence intervals for the regression coefficients). This can help to see how much the regional sensitivities discussed in this study are robust within the employed model framework.

The revised Figure 5 and the associated text are clearer than before. However, I am still having difficulty in understanding the method and result. So, what is exactly plotted in Figure 5? Is it $\beta_k^{region}$ from Equation (3) (page 6, line 30)? Or is it $\beta_k^{region}$ from another equation where the $P_{pre}^{region}$ in Equation (3) is replaced with the normalized $P_{pre}^{region}$? If the latter is true, then the "^" sign is missing in $P_{pre}^{region}$ in page 7, line 4.

Minor points

From the Article
Page 3, Line 33: what do the authors mean by "a previous run"?

From the Suppl. Material
Figure 2 y-axis labels: a typo in "Studenised"
Figure 3 caption: a typo in "hyperube"
Page2, line 1: a typo in "usedn"